# A Mechanistic Model for Cell Cycle Control in Which CDKs Act as Switches of Disordered Protein Phase Separation

**DOI:** 10.3390/cells11142189

**Published:** 2022-07-13

**Authors:** Liliana Krasinska, Daniel Fisher

**Affiliations:** IGMM, CNRS, INSERM, University of Montpellier, 34293 Montpellier, France

**Keywords:** cyclin-dependent kinase, intrinsic disorder, phase separation

## Abstract

Cyclin-dependent kinases (CDKs) are presumed to control the cell cycle by phosphorylating a large number of proteins involved in S-phase and mitosis, two mechanistically disparate biological processes. While the traditional qualitative model of CDK-mediated cell cycle control relies on differences in inherent substrate specificity between distinct CDK-cyclin complexes, they are largely dispensable according to the opposing quantitative model, which states that changes in the overall CDK activity level promote orderly progression through S-phase and mitosis. However, a mechanistic explanation for how such an activity can simultaneously regulate many distinct proteins is lacking. New evidence suggests that the CDK-dependent phosphorylation of ostensibly very diverse proteins might be achieved due to underlying similarity of phosphorylation sites and of the biochemical effects of their phosphorylation: they are preferentially located within intrinsically disordered regions of proteins that are components of membraneless organelles, and they regulate phase separation. Here, we review this evidence and suggest a mechanism for how a single enzyme’s activity can generate the dynamics required to remodel the cell at mitosis.

## 1. What Dictates CDK Substrate Specificity?

It is unclear how a single kinase’s activity can phosphorylate the plethora of CDK substrates [1,2,3,4] and coordinate such divergent processes as DNA replication, DNA repair, ciliogenesis, and mitosis, with the latter involving a complete cell reorganisation that happens in a matter of minutes, as well as controlling the epigenetic landscape (in embryonic stem cells). One possibility is that ostensibly diverse CDK substrates have underlying similarities, and their phosphorylation results in similar biochemical effects. A key aspect of such similarities that is usually considered involves the linear sequence surrounding the phosphorylation site. While early studies using peptides defined the optimal CDK1/2 consensus motif as S/T-P-X-K/R (where X corresponds to any amino acid) [5,6,7], the emerging picture is that most substrates do not conform to this optimum, and a significant minority do not have a proline following the phosphorylated S/T. By varying positions P +1 and P +3 (where P is the phosphorylation site) of a substrate and comparing crystal structures of CDK1 and CDK2, Endicott and colleagues showed that CDK1-cyclin B can readily target non-proline directed sites thanks to higher flexibility of the activation segment, which forms a platform for substrate recognition on both sides of the phosphorylated residue [8]. Indeed, the lack of absolute requirement for a proline in the +1 position has been largely confirmed by studying defined CDK substrates [9,10,11,12] or identifying CDK targets on a large scale [1,2,13]. For example, of the 19 mapped CDK1 sites on the budding yeast Wee1 homologue Swe1, 11 do not conform to the minimal consensus motif S/T-P [11]. Furthermore, in a recent phosphoproteomic analysis of CDK1 targets using chemical genetics in mouse embryonic stem cells, around 30–40% of phosphorylated serines and threonines were not followed by proline [1]. A new compilation of published human CDK sites found that, similarly to yeast [2], 10–20% are non-proline directed [13]. Finally, an in vitro study showed that sites with multiple K/R after the +1 position are more readily phosphorylated by CDK1 than S/T-P sites [14]. Nevertheless, despite the abundant evidence that CDKs do not absolutely require proline in the +1 position, most studies have excluded non-proline-directed sites from further analysis. Thus, the full landscape of CDK substrates is underestimated, and the consensus motif alone cannot explain how CDKs choose their targets.

Studies from the Loog lab suggested that the timing of a substrate’s phosphorylation (and thus ordering of the cell cycle) depends on differential substrate recognition by different CDK/cyclin complexes and is encoded in a substrate docking site in the cyclin, with a built-in mechanism of lower specificity of early CDK/cyclin complexes towards mitotic targets [15]. For multisite target phosphorylation, the Loog lab proposed a more complex set of features responsible for the choice of substrates, namely, the overall spatial pattern of the multisite region, distances between phosphorylation sites, distribution of target serines and threonines with respect to each other, the CKS1 (Cdc kinase subunit; small protein interacting with CDK1/2-cyclin complexes, a phospho-Thr acceptor and regulator of CDK function) phosphosite specificity, the positioning of cyclin-docking motifs, and the processivity at each step [16,17]. This model is largely based on processive phosphorylation, in which high-affinity sites are first phosphorylated, thus increasing affinity of the CDK-complex for the remaining low-affinity phosphosites due to phospho-binding of the CKS1 subunit. However, the evidence from Loog and colleagues rather suggests a distributive mode of multi-site substrate phosphorylation, i.e., depending on multiple transient interactions with the kinase, since it is subject to substrate inhibition [18]. Furthermore, while affinity differences of CDK1-cyclin complexes for different substrates clearly exist [15], and could in theory generate different timing of substrate phosphorylation in the cell cycle, this would rely on CDK-cyclin activity being strongly rate-limiting in vivo, yet the system appears to behave more like a bistable switch. In such a system, the protein phosphatase activity is more important in determining the substrate phosphorylation state (see accompanying review, [19]). Thus, it seems unlikely that a sophisticated code of substrate recognition could explain how hundreds of targets get phosphorylated by CDK1 at mitotic entry, and has not yet been confirmed in vivo. Indeed, analysis of substrates of different CDK-cyclin complexes [20] showed that intrinsic specificity towards any one substrate plays a minor role in choice of substrates. However, even if individual CDK-cyclin complexes are not strictly essential for phosphorylations that occur early in the cell cycle, cyclins expressed in S-phase probably evolved to have higher specific activity against these targets.

Specificity is also dictated by where a particular substrate is present inside the cell and when a given complex is present and active. This was illustrated by the study of Pagliuca and colleagues [21], which analysed, using mass spectrometry, proteins interacting in vivo with cyclin E, A and B in different stages of the cell cycle. This showed that many substrates bind to different cyclins, but that this was dictated by a cell cycle phase, confirming that targeting of a protein by any one CDK/cyclin complex is context-dependent. These conclusions are consistent with results obtained by other labs and in different systems, which elegantly demonstrated that no cyclin or CDK is indispensable, and instead, it is the overall kinase-to-phosphatase activity ratio that dictates in which phase of the cell cycle a cell is [22,23]. Thus, in the absence of CDK2, S-phase can be controlled by CDK1 [24,25,26], and if overall kinase activity is sufficiently high, CDK1 and cyclin B can be substituted by CDK2 and other cyclins to trigger entry into mitosis [27,28] (see also [19] in the same issue).

## 2. Structural Effects of Protein Phosphorylation

To understand this redundancy, it is important to consider the general biochemical and biophysical effects of protein phosphorylation and the structural nature of substrates. Analyses of CDK1 phosphosites in budding yeast [2,29] provided several important insights into their shared features. First, around 90% are located on predicted loops or intrinsically disordered regions (IDRs), and such sites are often clustered in the same protein; this is also true for mouse embryonic stem cells [1]. Moreover, analysis of their conservation and sequence context across 32 fungal species revealed that the sites evolve differently depending on their localisation [2]. The position of sites located in structured regions is usually strictly conserved, suggesting that phosphorylation causes precise conformational changes that are functionally important. However, most sites, which are located in disordered regions, are conserved only in the very closely related species, yet the same regions tend to be phosphorylated in other positions in many species. Indeed, disordered regions evolve differently from structured ones, accepting more deletions and insertions [30], but they are still under strong selection: if evolutionary models derived from ordered proteins are applied to IDPs, the disorder is lost, implying that maintaining a disordered region during evolution is more difficult, and random mutations easily destroy these features [31]. Weakly conserved phosphorylations were long considered to be functionally irrelevant. However, recent studies have demonstrated their importance for protein–protein and protein–DNA interactions. This can readily be explained by phosphorylations regulating bulk electrostatic charge of protein surfaces or producing binding sites for phospho-binding domains (such as SH2, FHA, WD40 or 14-3-3) [32], where precise positioning of the phosphorylation is not required. Intrinsic disorder provides plasticity that supports multivalent low affinity interactions, allowing intrinsically disordered proteins to serve as hubs for binding many different partners [33,34]. Functions of disordered proteins are related to the conformations they can adopt. Advanced theoretical simulations compute inter-residue distance profiles, between any two residues, that provide a detailed description of the ensemble of conformational states of a given IDP [35,36]. Most IDRs are polyampholytes, containing both positively and negatively charged residues. The fraction of charged residues and the linear sequence distribution of oppositely charged amino acids dictate the conformation of IDRs, which can be theoretically predicted [35,37]. Weak polyampholytes form globules. When the fraction of charged residues is higher, in well-mixed polyampholytes, the intrachain repulsions and attractions are counterbalanced and the polymers are described as either performing ‘random walks’ in three dimensions or existing as ‘random coils’ where monomers are randomly spaces while bound to adjacent units. When oppositely charged residues are segregated within the sequence, hairpin-like structures form, induced by long-ranged electrostatic attractions [37]. Phosphorylation can regulate the function of IDRs in various ways (Figure 1A). Addition of phosphate groups adds negative charge that can change a polyelectrolyte into a polyampholyte or alter charge distribution within the protein, thus affecting its conformation; specific hotspots for modifications are observed that can significantly alter IDR density profiles [35]. Phosphate groups also permit formation of hydrogen bonds and salt bridges, that can impact on conformation and dynamics of IDRs [38,39,40] and thus on their interaction with partners (e.g., [41,42]). Particular biophysical features of IDRs provide them with the ability to undergo induced folding or unfolding, and to acquire different structures upon binding with different partners, resulting in gain or loss of function [43]. Phosphorylation can trigger any of these transitions (Figure 1A). For example, the translational repressor eukaryotic initiation factor (eIF) 4E-binding protein 2 (4E-BP2) specifically and tightly binds to eIF4E in its unphosphorylated state, and, upon phosphorylation, it undergoes a disorder-to-order transition which significantly weakens its interaction with eIF4E, at which point it gets outcompeted for eIF4E binding by eIF4G, thus triggering initiation of translation [42]. Another well-studied example is the intrinsically disordered regulatory (R) region of the cystic fibrosis transmembrane conductance regulator (CFTR). It is phosphorylated on over ten sites by different kinases (protein kinase A, protein kinase C, and 5′-AMP-activated protein kinase) and these modifications extensively alter the protein function and interactions [44,45]. In an unphosphorylated state, the R region interacts with nucleotide binding domains, NBD1 and NBD2, of CFTR, precluding opening of the ion channel [46]. Upon phosphorylation, although no disorder-to-order transition takes place, the R region becomes even less compact and its interactions change drastically [44]. It disengages from NBD1 and NBD2 [47] and becomes involved in different homo- and heterotypic interactions, with some of them happening simultaneously, thanks to the huge flexibility and moderate affinity of binding of the R region, resulting in opening of the ion channel [44].

That phosphorylations or other post-translational modifications (including acetylation, methylation, hydroxylation, sulfotyrosination, and others) preferentially locate to intrinsically disordered regions (IDRs) of proteins is not a new concept [48]. In 2004, Iakoucheva and colleagues [49] analysed properties of amino acids surrounding over 1500 experimentally confirmed eukaryotic phosphosites. Amino acid composition, hydrophobicity, flexibility, charge, and other properties related to the phosphorylated residues were found to be similar to the characteristics of intrinsically disordered regions. They were enriched in flexible and surface-exposed amino acids: arginine, lysine, serine, proline, and glutamic acid, and depleted of hydrophobic, rigid and neutral ones, such as cysteine, phenylalanine, tryptophan, valine, isoleucine, leucine, and asparagine. Moreover, the study found that ten times more phosphorylation sites are located in predicted disordered proteins than in structured proteins, which indicates the importance of low-affinity and high-specificity interactions between kinases and their substrates.

## 3. CDKs Mainly Phosphorylate Disordered Proteins of Membraneless Organelles

Several questions emerge from these results. Are CDKs distinct or similar in this respect to other kinases? And what about other cell cycle kinases? Or MAPK kinases, which share with CDKs the proline-directed consensus site, but regulate signalling pathways rather than cell cycle processes? Is the preferential phosphorylation of IDRs by CDKs a conserved phenomenon? Is the observed enrichment of phosphorylations in disordered regions a simple consequence of the compositional bias whereby serines and threonines, as well as prolines, are in general enriched in intrinsically disordered regions? And are the biochemical and biophysical effects of CDK-mediated phosphorylations on diverse substrates similar? Recent work by Valverde, Dubra and colleagues addressed these exact questions [13]. 

To do so, they analysed CDK1/2 targets in human and budding yeast, and cell cycle phosphorylation dynamics using *Xenopus laevis*, whose early embryonic divisions are naturally synchronous—a system devoid of artifacts due to cell synchronisation approaches. High-resolution mass spectrometry analysis of phosphorylations throughout early divisions of single *X*. *laevis* embryos as well as during the DNA replication time course and in mitosis in *Xenopus* egg extracts provided an unbiased map of dynamic cell cycle phosphorylations. A quarter of proteins with such dynamic sites are homologues of known human CDK targets, while motif analysis revealed that more than half of the dynamic phosphosites comply with at least the minimum CDK consensus motif, underlining the predominance of CDK phosphorylations in the regulation of the cell cycle. Correction for compositional bias established that, indeed, despite a higher number of serines, threonines and tyrosines located to IDRs, CDKs preferentially target sites in disordered versus structured regions, both in yeast and humans. The same was true for Plk1, Aurora kinases, DYRK, NEK and MAPK sites, and thus is not specific to CDKs. However, it emerged that human and yeast CDK targets, as well as *Xenopus* dynamic cell cycle phosphoproteins are twice as disordered on average as other phosphoproteins (Figure 2). This indicates the importance of intrinsic disorder for the regulation of cell cycle processes, and was corroborated by the fact that substrates of other cell cycle kinases, but not MAPK (that share the same consensus motif as CDK but control cell signalling), conformed to the same principle. Thus, cell cycle kinases in general, and CDKs in particular, appear to have been selected for their ability to phosphorylate disordered regions with limited sequence specificity.

Why might this be so? Intrinsically disordered proteins have recently been recognised for their role in driving phase separation. Liquid–liquid phase separation (LLPS) allows cells to organise molecules (proteins, RNA and DNA) into three-dimensional compartments, so-called membrane-less organelles (MLOs), that are found both in the nucleus (nucleoli, nuclear pore complexes, Cajal bodies, splicing speckles, PML bodies and 53BP1 bodies) and in the cytoplasm (including stress granules and P bodies) and participate in a variety of biological functions, such as RNA metabolism, transcription, nucleocytoplasmic transport, DNA damage, stress responses and others [50,51,52]. Many of these MLOs are regulated throughout the cell cycle. The LLPS-promoting ability of IDRs resides in their multivalency, the ability to interact, with low affinity but high specificity, and thus dynamically, with many partners, forming very dense networks of interactions that drive phase separation [53]. A model of ‘stickers and spacers’ describes these interactions, where stickers link neighbouring protein chains (or RNA molecules) through non-covalent weak interactions, and spacers provide flexibility [54]. Structured proteins possess few such stickers, which are constituted by binding patches. However, in IDRs, that can adopt an ensemble of conformational states, stickers can be single residues or short linear motifs, or the combination of the two. Given the importance of intra- and intermolecular interactions in phase separation, reversible post-translational modifications are employed to regulate the MLO formation, composition and dynamics [53]. Phosphorylation has also been shown to affect the association and dissociation of particular components, dissolution of whole phase-separated compartments, and their dynamics [55,56,57,58,59,60,61]. 

This applies to CDKs. At the entry to mitosis, CDK1 phosphorylates many nucleoporins (NUPs; [4]). NUPs form nuclear pores, structures responsible for selective bidirectional transport of molecules between the nucleus and the cytoplasm. A third of around thirty or so NUPs, those lining the interior of the nuclear pore, possess FG (phenylalanine-glycine)-rich disordered regions that were demonstrated to undergo LLPS, forming hydrogels that mimic the permeability barrier of the nuclear pore [62,63], the maintenance of which depends on NUP98 [64,65]. The phosphorylation of the FG repeat-rich N-terminus of NUP98 by CDK1 was shown to be crucial for nuclear pore disassembly [66], a key event for nuclear disintegration in mitosis. In budding yeast, CDK1 (Cdc28) was also demonstrated to regulate the stability of stress granules (SGs, [67]), cytoplasmic condensates that appear in response to various exogenous stress and accumulate untranslating mRNAs and RNA-binding proteins [68]. An RNA-binding protein, Whi8, was found to interact with mRNA of G1 cyclin Cln3 and sequester it in SGs, thus downregulating Cdc28 activity under stress conditions. The same protein interaction with Cdc28 was responsible for bringing the kinase to SGs and, through phosphorylation of many targets, dissolving these structures when cells returned to non-stress conditions. This mutual inhibition was demonstrated to behave as a bistable system [67]. CDKs might similarly regulate SG stability in mammalian cells, as a screen for inhibitors of SG dissolution during recovery from stress in HeLa and U2OS cells identified, among others, to be CDK2 and CDK4 inhibitors [59]. One other interesting example is the CDK-mediated regulation of phase separation of replication initiation factors. Fruit fly Orc1, Cdc6 and Cdt1 all possess intrinsically disordered regions and can phase separate in vitro, in the presence of DNA [69,70]; the same was shown for human ORC1 and CDC6 [71]. Multiple CDK phosphorylation sites are located in IDRs of these proteins and CDK-mediated phosphorylation abolishes their phase separation [69]. Replication foci have not yet been shown to have the characteristics of phase-separated condensates in vivo, though such a hypothesis is highly plausible; indeed, high CDK activity inhibits DNA replication and blocks association of replication factors with chromatin [23,72,73,74,75], which might in part be due to dissolution of replication foci. 

The examples described above demonstrate that CDKs can regulate the organisation and assembly of specific MLOs. However, given the predominance of CDK-mediated sites among cell cycle phosphorylations, the general high intrinsic disorder of CDK substrates [13], and the switch-like reorganisation of the cell at mitotic entry when most of the biological condensates dissolve, one might hypothesise that CDKs evolved to phosphorylate IDRs and regulate phase separation during the cell cycle (Figure 1B). So far, this has not been demonstrated. Indeed, a different kinase, DYRK3, has been suggested to control dissolution of certain liquid organelles in mitosis [61]. DYRK3 inhibition led to formation of aberrant hybrid condensates in mitotic cells, that stained for components of splicing speckles, stress-granules and pericentriolar material. Thus, in the absence of DYRK3 kinase activity new structures are formed, but how existing MLOs dissolve at the entry into mitosis remains unknown. No aberrant condensation of factors of other tested organelles, including nucleoli, Cajal bodies, or P-bodies was observed upon loss of DYRK3 activity, suggesting that these MLOs are regulated by a different mechanism. Furthermore, analysis of DYRK3 activity level showed that it increased throughout S phase, reaching its maximum in G2, without any further increase in mitosis; yet, no dissolution of splicing speckles is observed in G2. We propose that another kinase, namely CDK1, might contribute to disassembly of physiological phase-separated organelles upon mitotic entry (Figure 1B).

## 4. CDK-Mediated Phosphorylation Can Promote or Inhibit Phase Separation

To test the hypothesis that CDK might act as general MLO dissolvase, Valverde, Dubra and colleagues [13] assembled a human MLO proteome and found that over one third of these proteins are CDK targets. These included major factors of the analysed condensates, such as coilin of Cajal bodies, numerous nucleoporins (NUP53, NUP98, NUP153, ELYS, and others); nucleolin and nucleophosmin of nucleoli; 53BP1, RIF1 and MDC1 of 53BP1 bodies; promyelocytic leukaemia protein of PML bodies; and MED1, which drives phase separation of the transcriptional apparatus [76]. Analytical modelling of a subset of IDPs specific to several MLOs, that accounts for the effects of sequence-specific interactions on chain conformation, which dictate the distance between any two amino acid residues, and which correlates with the propensity to phase separate [35] showed that CDK-mediated phosphorylation is a key regulator of homotypic interactions and is likely to affect phase separation. To better understand the effect of CDK phosphorylation, the authors concentrated on a model intrinsically disordered target, the cell proliferation marker Ki-67. It is an almost entirely disordered protein, with the major part constituted by multivalent Ki-67 repeats of unknown function and heavily phosphorylated by CDK1 in mitosis [77,78]. While in interphase Ki-67 localises to nucleoli and organises heterochromatin [79], its best-established function is to allow formation of the perichromosomal layer in mitosis [80,81], an entity that provides physical separation of chromosomes and that is likely to be phase-separated [78]. Both analytical modelling and coarse-grained molecular dynamics simulations revealed that CDK phosphorylation of the full-length Ki-67 should increase its ability to phase separate, while the opposite was true for the single consensus repeat of Ki-67 [13]. The predictions of models were confirmed by experiments in cells and in vitro. Using an optogenetic system [82], the full-length Ki-67 phase-separated in cells, a phenomenon that was accentuated when the CDK-opposing phosphatase was inhibited, but was abolished upon CDK inhibition. On the other hand, purified Ki-67 consensus repeat phase separated in vitro, which was prevented by full phosphorylation by CDK. Together with the fact that CDK-mediated phosphorylation increases mobility of, and reduces affinity between, nucleolar components [83], and decreases phase separation propensity of nucleolar factors (as shown by analytical modelling for nucleolin and nucleophosmin; [13]), an interesting hypothesis emerges in which CDK1 triggers dissolution of nucleoli at mitotic entry and phosphorylated Ki-67 acts as a scaffold for perichromosomal layer assembly (Figure 1B). 

These results are consistent with another recent study [84], based on the theory that predicts that for a polyampholytic IDR with zero net charge, the propensity to phase separate is correlated with “blockiness” (see above; [85]). They concentrated their investigation on one of the repeats of Ki-67 that showed the most predominant “blockiness” when mitotically phosphorylated: division of the sequence into positively and negatively charged blocks. In in vitro experiments, the recombinant repeat weakly phase separated, and the droplet formation was increased with the number of repeats, by CDK1-mediated phosphorylation, or for the phospho-mimic mutant or a mutant with accentuated blocky charge pattern. Further studies; however, will be required to ascertain the effects of CDK1-mediated phosphorylation in vivo. It will also be important to build more complex models, including Ki-67 interacting partners and constituents of MLOs, to understand how heterotypic interactions are regulated by CDK-dependent phosphorylation and with what effect on phase separation.

## 5. Conclusions

The existing data thus provide evidence that CDKs have a much less constrained choice of targets than previously thought, and they differ from other phosphoproteins in that they are more disordered and a large fraction of them localise to MLOs. CDK-mediated phosphorylation appears to act as a key regulator of homotypic interactions of IDRs, and it can either favour or inhibit phase separation. Consequently, changes in the ratio of the overall CDK to phosphatase activity can provide a general mechanism for initiation of various processes during the cell cycle, with its highest level triggering a switch-like cell reorganisation in mitosis and disassembly of most phase-separated condensates. There is therefore no need to invoke a highly complex code of docking motifs and phosphosites on the substrate, since the overall phosphorylation of multiple sites modifies protein–protein interactions, and the absolute kinetics of each phosphorylation is probably not critical. Furthermore, since there is double negative feedback between the CDK and its opposing phosphatase, a small relative increase in the overall CDK activity can trigger a complete change in substrate phosphorylation. This is consistent with recent data (see above) on the synchrony of mitotic phosphorylation in which no differences in timing of phosphorylation of highly diverse substrates are observed. Our model also provides a mechanistic explanation for how a single CDK-cyclin’s activity can drive a fairly normal cell cycle, and it suggests that CDKs were selected over evolutionary timescales to be rather un-specific kinases that are very efficient at phosphorylating disordered proteins, since much of the reorganisation that occurs in a dividing cell relies on such proteins. Other mitotic kinases (e.g., polo-like kinases) may have evolved to act in a similar manner but targeting IDRs that have more acidic regions around the phosphorylation site, that CDKs cannot readily phosphorylate.

## Figures and Tables

**Figure 1 cells-11-02189-f001:**
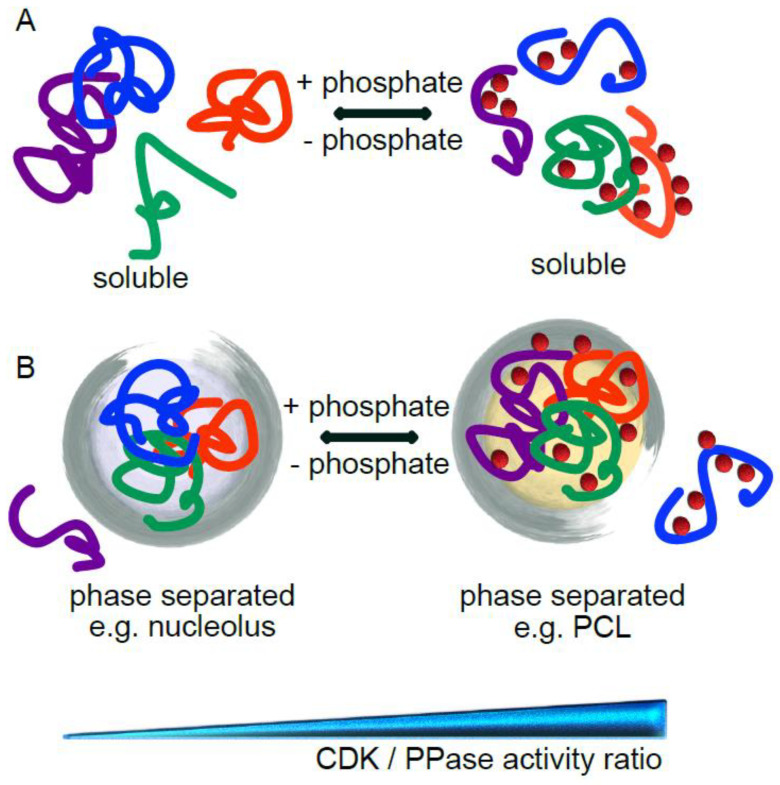
Phosphorylation controls the structure and function of IDRs in various ways. (**A**) Phosphorylation can alter IDR conformation, inducing either folding or unfolding, thus affecting protein interactions. (**B**) CDK-mediated cell cycle-regulated phosphorylation acts as “dissolvase”, leading to dissolution of various MLOs (e.g., nucleoli) at the entry into mitosis, while inducing formation of other phase-separated compartments (e.g., perichromosomal layer, PCL), by regulating homotypic and heterotypic IDR interactions. Red circles indicate phosphorylation.

**Figure 2 cells-11-02189-f002:**
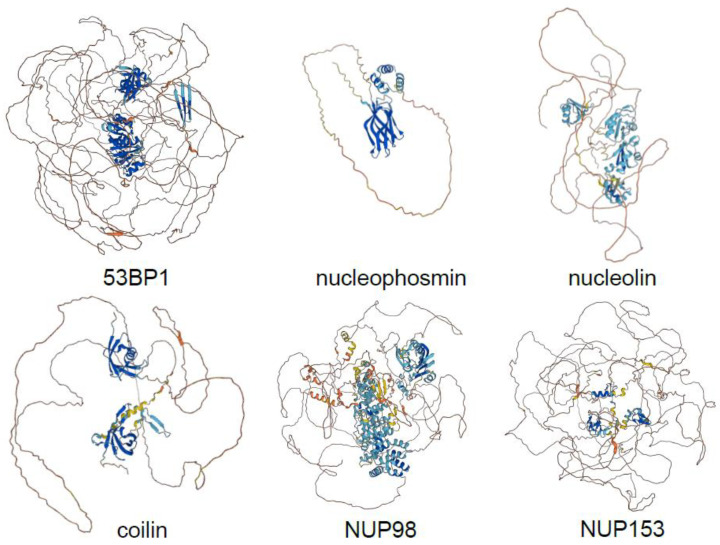
AlphaFold-predicted structures of exemplary IDPs and prominent MLO components that are CDK substrates (https://alphafold.ebi.ac.uk/) (accessed on 1 June 2022).

## Data Availability

Not applicable.

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
