# Peer review of "A Mechanistic Model for Cell Cycle Control in Which CDKs Act as Switches of Disordered Protein Phase Separation"

_cells, 2022, doi:10.3390/cells11142189_

Round 1
Reviewer 1 Report
The manuscript by Krasinska and Fisher discuss a hypothesis that phosphorylation by Cdk leads to a change in phase separation of the target, at least some of them.
Traditionally, it is assumed that phosphorylation leads to a specific change in conformation or interactions due to additional negative charges. For the case of the central regulator of the cell cycle the model is discussed that the actual sites of physophorylation are not important. In a large fraction of targets regions of the protein are phosphorylated with intrinsically disordered structure and no consensus sequence. It is then hypothesized that this phosphorylation may lead to phase separation (and formation of so-called “membrane-less organelles”) or vice versa. Phase-separated granules contain a certain fraction of intrinsically disordered proteins.
This model is then applied to the central regulator of the cell cycle, Cdk-cyclin complexes. It is a long-standing and unresolved problem in the cell cycle field, that a single Cdk-Cyclin complex can control entry into S phase with a distinct class of targets and later into M phase with another set of targets. For example, proteins involved in DNA replication are activated in S phase and mitotic proteins such as components of the nuclear envelope and chromatin are phosphorylated in M phase. Although this has been most clearly shown in yeast, in higher organism this notion also holds true, as Cdk1 can substitute Cdk2 for control of S phase and Cdk2 can substitute Cdk1 for control of M phase.
As an explanation for this redundancy, the dominant view is that low Cdk activity leads to S phase and high activity to M phase. Although there are some attractive features of this dominant model, it leads to complications, which are often simply neglected.
Having lengthy written this, the manuscript by Krasinska provides a fresh view in relation to the dominant model. The literature is as much as I see carefully and without bias incorporated. The weaknesses of the dominant model are properly named and not overstated. The issue of sequence-specific and “unspecific” target sites is properly presented and supported by ample literature references.
The two figures suitable.
There a few small spelling errors. I have only one request for revision. The Abstract could be improved. The features and also limitations of the quantitative (“dominant”) model should be more clearly stated. The sentences in line 13 and following are not clear to me. Similarly, the description of the alternative model, involving “unspecific” phosphor sites and potentially phase transitions in lines 16 and following should be more clearly formulated. For example the term “phase separation” is not mentioned in the abstract although it is in the title and a central aspect of the model.
I had read the abstract several time to get to the point. And it was only after having read the full manuscript that I understood the new ideas.
Reviewer 2 Report
This review by Krasinska and Fisher discusses a model whereby Cyclin-Dependent Kinases control the cell cycle by preferentially targeting disordered regions in substrate proteins that tend to undergo phase separation. This piece is largely an extension of an unpublished manuscript posted on a preprint site and of which Krasinska and Fisher are co-authors (Altelaar et al, Reference 13 in the current manuscript).
As a cell cycle expert with a focus on kinases and phosphatases, I find this manuscript very interesting. I have learned a lot while reading it and it made me think in new ways. The text is very well written. I strongly support publication after consideration of the few following points.
I understand that the authors are pushing their model here, but I feel like a bit more room and credit should be given to alternative models. In particular, the Loog models where Cyclin-CDKs follow a more precise sequence code in targeting their substrates should not be completely discredited. Similarly, although much evidence supports the idea that global the CDK activity level largely determines substrate phosphorylation and cell cycle transitions, there is also evidence for substrate specificity provided by cyclins, including from budding yeast (for example, see Bloom & Cross. Nat Rev Mol Cell Biol, 2007). As in many cases in biology, the actual reality here is almost surely a mix of these different models.
I find that Figure 2 could be improved by indicating known phosphorylation sites with red circles in the modeled proteins.
Reviewer 3 Report
I like the review as it provides a noncommon viewpoint on the key cellular process - cell cycle regulation. I would call this more a hypothesis than a review, but still it looks very interesting and fresh for readers of Cells journal. The main drawback - the manuscript is over positive in terms of supporting the hypothesis presented by authors, so I suggest to add some cons. For example, classic checkpoints still looks functional and their role should be implemented in the general picture presented here. Cell cycle is too complex to be explained by a single mechanism. I appreciate consistency of the authors, but I suggest to implement classic counterparts of cycle driving. Also I suggest to improve figures (Fig.1: Ph - is not appropriate for phosphate - the full word will be more common for readers, Fig. 2: I suggest to show, how phosphorylation can influence on the ordering/disordering of the presented proteins).
The paper is interesting and I suggest minor revision before it can be accepted by Cells journal
